# Visual Clues: Bridging Vision and Language Foundations for Image Paragraph Captioning

**Yujia Xie, Luowei Zhou***, **Xiyang Dai, Lu Yuan, Nguyen Bach, Ce Liu, Michael Zeng**
Microsoft
{yujiaxie, luowei.zhou, xiyang.dai, luyuan, nguyenbach, ce.liu, nzeng}microsoft.com

## Abstract

People say, "*A picture is worth a thousand words*". Then how can we get the rich information out of the image? We argue that by using *visual clues* to bridge large pretrained vision foundation models and language models, we can do so without any extra cross-modal training. Thanks to the strong zero-shot capability of foundation models, we start by constructing a rich semantic representation of the image (*e.g.*, image tags, object attributes / locations, captions) as a structured textual prompt, called *visual clues*, using a vision foundation model. Based on visual clues, we use large language model to produce a series of comprehensive descriptions for the visual content, which is then verified by the vision model again to select the candidate that aligns best with the image. We evaluate the quality of generated descriptions by quantitative and qualitative measurement. The results demonstrate the effectiveness of such a structured semantic representation.

## 1   Introduction

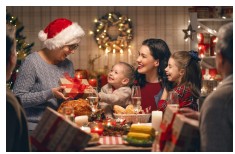 *This image is of a family celebrating Christmas. They are all gathered around a dinner table, with a turkey and other food on it. The family is smiling and seems to be enjoying themselves. There is a Christmas tree in the background and some Christmas lights on the walls.*

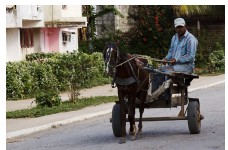 *This image features a horse and buggy travelling down a road in the town of Holguin, Cuba. The horse is harnessed to the buggy and is pulling it along, while a farmworker rides behind. This image captures the everyday life of Cubans, with their traditional horse-drawn carts still in use.*

Figure 1: Examples of generated image paragraph.

"*Vision is a process that produces from images of the external world a **description** that is useful to the viewer and not cluttered with irrelevant information.*"

—David Marr, *Vision, p31*

What makes a good "*description*" for vision? Over the past several decades, computer vision pioneers drew inspiration from neural science, cognitive science, and psychophysics (Marr, 2010), pointing us to the North Stars (Fei-Fei and Krishna, 2022), some among them being image classification and object detection. Despite the tremendous progress that has been made, much of these object-centric works remain a proxy for an eventual task or application that requires a holistic view of the visual content, involving concepts beyond objects: actions, attributes, and relations, to name a few.

---

*Currently at Google Brain.

36th Conference on Neural Information Processing Systems (NeurIPS 2022).

In our work, we argue that textual representation suffices such "description". It brings forth a more holistic visual representation than categorical labels. It allows machines to interpret visual signals through descriptive captions (Zhou et al., 2020b; Li et al., 2022), and perform more language-heavy tasks such as question-answers (Rajpurkar et al., 2016), or multi-round dialogues (Li et al., 2017). On the other hand, the access to abundant web multimodal language data (*e.g.*, image alt-text, video subtitles) provides us with the fuel for powering neural visual representations from contrastive language-image pre-training (CLIP, Yuan et al. (2021); Radford et al. (2021)). The marriage of the two renders a new computer vision system that is faithful, generic, and versatile.

We name this new computer vision system **BEST**, for Bridging with Explicit Structured Textural clues. We start by constructing a semantic representation of the image. This semantic representation, which we referred to as *visual clues*, comprises rich semantic components, from object and attribute tags to localized detection regions and region captions. Powered by the recent advances in vision foundation model Florence (Yuan et al., 2021), the visual clues are rich in open-vocabulary expressions, marking a major difference compared to existing symbolic approaches (*e.g.*, scene graphs Krishna et al. (2017)) with closed-set vocabularies.

The visual clues are interpretable, not only for humans, but also for machines. Take the generative language model GPT-3 (Brown et al., 2020). The visual clues could be digested by GPT-3, which in return produces crisp language descriptions that are sensible to the viewer while not cluttered with irrelevant information from the visual clues. Whereas this open-loop process could potentially suffer from object hallucination issues (Maynez et al., 2020; Zhou et al., 2020a) as the outputs from GPT-3 are not governed by any means, we further deploy a closed-loop verification procedure that grounds descriptions back to the original image.

To evaluate the quality of the language descriptions, we resort to an existing task named Image Paragraph Captioning (IPC), but with a twist. IPC aims to address the demand for generating long, coherent, and informative descriptions of the whole visual content of an image (Krause et al., 2017), which can eventually be used for many applications including poetry composition (Liu et al., 2018), automatic recipe generation (Salvador et al., 2019), visual storytelling (Huang et al., 2016), advertisement generation, or help blind or visually-impaired people see better. The existing metrics for IPC such as BLEU (Papineni et al., 2002), METEOR (Denkowski and Lavie, 2014), and CIDEr (Vedantam et al., 2015) encourage exact matching between semantics in generated captions and those in the reference. However, they over penalize visual details that are not annotated thus compromising their qualifications for measuring overall representation quality. Inspired by Anderson et al. (2016); Krishna et al. (2017), we propose to measure the accuracy on *scene graphs* extracted from generated text against human-annotated graphs, which, as suggested by Anderson et al. (2016), co-relates better with human judgment.

The contributions are twofold. First, we propose a general framework for semantic visual representation and showcase its application to image paragraph captioning. The framework is simple yet highly extendable, allowing new components to be plug-in and supporting other use scenarios that require a holistic view of the visual content. Second, we benchmark the effectiveness of the proposed model on its capacity for representing visual concepts (*e.g.*, scene graphs) and set new state-of-the-art results.

**Notations.** We denote $\langle \cdot, \cdot \rangle$ as inner product between two vectors, $|\mathcal{A}|$ as the cardinality of set $\mathcal{A}$.

## 2 Related Works

**Image paragraph generation.** The task of generating image paragraphs is first introduced by Krause et al. (2017). Conditioned on the visual features, they first train a sentence recurrent neural network (RNN) to output sentence topics, and then feed each of the topics into another RNN to generate the paragraphs. Liang et al. (2017) further improve the hierarchical RNN framework by introducing an adversarial discriminator for smoother transitions between sentences. Chatterjee and Schwing (2018) also address cross-sentence topic consistency by a global coherence vector. Melas-Kyriazi et al. (2018) add a repeat penalty to the optimization, to prevent the appearance of repeated sequences. Wang et al. (2019) use convolutional auto-encoder for topic modeling based on region-level image features. Along this line, many other works have been done (Dai et al., 2017; Luo et al., 2019; Mao et al., 2018; Xu et al., 2020; Guo et al., 2021; Shi et al., 2021). Most of the proposed models, however, are trained on *Stanford image-paragraph dataset* (Krause et al., 2017), which only contains 14 thousand of training paragraphs for its expensive nature to collect. Due to lack of data, the generated paragraphs usually lack coherence both locally and globally. Therefore, many of the above works aim to make the best use of data to improve the coherence. Yet nowadays,

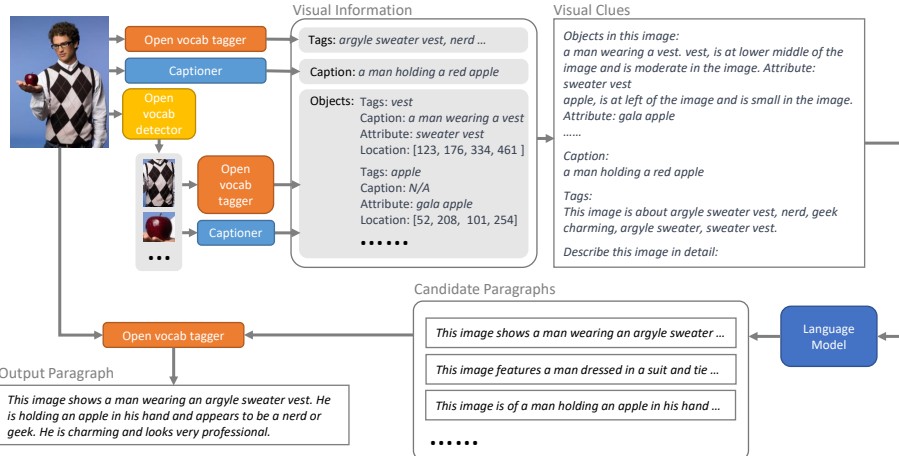

Figure 2: Framework demonstration. The orange *open vocab tagger* box corresponds to the image encoder $f_v(\cdot)$ and the text encoder as $f_t(\cdot)$. The blue *captioner* box is the caption model $c(\cdot)$. large language models can generate long coherent paragraphs by default. Our work, leveraging recent progress of large pretrained models, focuses more on how to guide and constrain the generated text instead.

**Constrained text generation** In recent years, rapid progress has been made in vision-language pretraining (VLP). CLIP (Radford et al., 2021), ALIGN (Jia et al., 2021), and Florence (Yuan et al., 2021) are proposed to encode vision and language into a *joint* representation space for crossmodal alignment tasks, *e.g.*, zero-shot image classification. Another line of research, *e.g.*, SimVLM (Wang et al., 2021), FLAVA (Singh et al., 2021), BLIP (Li et al., 2022), CoCa (Yu et al., 2022) and many others (Cho et al., 2021; Wang et al., 2022; Zhu et al., 2021; Alayrac et al., 2022) adopt encoder-decoder models trained with generative losses. Those models are capable of performing image captioning in a zero-shot manner. A concurrent work, Socratic Models (SM, Zeng et al. (2022)), also use textual data to bridge the domain gap between vision-language models and language models. The model, however, is stronger in retrieval tasks than captioning tasks as we will show later. There are also other works leveraging large language models to solve vision tasks, *e.g.*, PICa (Yang et al., 2021) uses GPT-3 (Brown et al., 2020) to extract commonsense knowledge for visual question answering tasks, MAGIC (Su et al., 2022) uses a CLIP-induced score to regularize the language generation so that it is semantically related to the given image, and VisualGPT (Chen et al., 2022) employs a self-resurrecting encoder-decoder attention mechanism to adapt the language models with a small amount of in-domain image-text data.

## 3 Framework

Given an image $I$, our goal is to generate long and coherent descriptive text based on image inputs, leveraging only the existing pretrained models. Our framework can be divided into three stages:

1. Represent $I$ with visual clues $S$, which contain the rich visual information;
2. Feed the visual clues into a language model to generate $K$ candidate paragraphs $\{T_i\}_{i=1}^K$;
3. Select the best paragraph $T^*$ from the candidates $\{T_i\}_{i=1}^K$.

The overall framework is illustrated in Figure 2. We will then elaborate on each of them.

### 3.1 Visual Clue Extraction

We leverage three state-of-the-art models with the open-vocabulary capability to extract the visual information, namely, the concise tags, the short captions, and the local descriptions.

**Concise tags.** The first model we use is the contrastively trained vision-language models, *e.g.*, CLIP (Radford et al., 2021), Florence (Yuan et al., 2021). Such models are pretrained on image-text pairs $\{x_i, y_i\}$, and is composed of the image encoder $f_v(\cdot)$ and the text encoder $f_t(\cdot)$. Given a minibatch $\mathcal{B}$, the models are optimized by contrastive loss

$$\mathcal{L} = -\frac{1}{|\mathcal{B}|} \sum_{x_i, y_i \in \mathcal{B}} \left( \frac{\exp(\langle f_v(x_i), f_t(y_i)\rangle/\tau)}{\sum_{y_j \in \mathcal{B}, j \neq i} \exp(\langle f_v(x_i), f_t(y_j)\rangle/\tau)} + \frac{\exp(\langle f_v(x_i), f_t(y_i)\rangle/\tau)}{\sum_{x_j \in \mathcal{B}, j \neq i} \exp(\langle f_v(x_j), f_t(y_i)\rangle/\tau)} \right),$$

where $\tau$ is the temperature. This loss explicitly uses inner product $\langle \cdot, \cdot \rangle$ to measure the similarity between the encoded image $f_v(x_i)$ and encoded text $f_t(y_j)$, and higher similarities are encouraged if the images and texts are paired. Therefore, such a pretrained model is capable of selecting the tags that describe the image $I$ from a set of customized tags by computing the similarities. Given a set of tags $\{t_i\}_{i=1}^N$, we compute the similarities between the input image $I$ and the tags, and adopt the tags with top-$M$ similarities,

$$\mathcal{T} = \{t_j^*\}_{j=1}^M = \underset{t_i, i=1,\cdots,N}{\arg \text{top-M}} \langle f_v(I), f_t(t_i) \rangle. \tag{1}$$

**Short captions.** The second model is a caption model $c(\cdot)$. We use it to generate an overall image description $c(I)$.

**Local descriptions.** The third model is an object detection model. We adopt a well-trained object detector, to provide us with the locations of the possible objects in the format of bounding boxes. The bounding boxes are processed with the non-maximum suppression technique to filter out repetitions. Denote the object proposals as $\{b_j\}_{j=1}^R$ and image regions cropped from corresponding boxes as $\{p_j\}_{j=1}^R$. We first select the indices of the bounding boxes with objects that can be named by our customized tag set,

$$\mathcal{P} = \{\ell_k\}_{k=1}^Q = \{j | \langle f_v(p_j), f_t(t_i) \rangle > \beta, i = 1, \cdots, N, j = 1, \cdots, R\}. \tag{2}$$

Here, $\beta$ is a threshold certifying whether $t_i$ is aligned with $p_j$. Given a set of customized attribute $\{a_i\}_{i=1}^V$, each selected proposal $\ell_k$ from $\mathcal{P}$ is then assigned to an attribute

$$a_{\ell_k}^* = \underset{a_i, i=1,\cdots,V}{\arg\max} \langle f_v(p_{\ell_k}), f_t(a_i) \rangle, \tag{3}$$

and the corresponding tags

$$\mathcal{O}_{\ell_k} = \{t_i | \langle f_v(p_{\ell_k}), f_t(t_i) \rangle > \beta, i = 1, \cdots, N\}. \tag{4}$$

In addition to the tags and attributes to the bounding boxes, we also use the caption model $c(\cdot)$ to provide some more descriptive texts $\{c(p_{\ell_k})\}_{k=1}^Q$.

In summary, we collect a tag set $\mathcal{T}$ and a caption $c(I)$ as global descriptions to the image, and a quadruple $(b_{\ell_k}, a_{\ell_k}^*, \mathcal{O}_{\ell_k}, c(p_{\ell_k}))$ as local descriptions for each selected bounding box.

### 3.2 Candidate Synthesis

We then format the collected visual information into the structured *visual clues*, which can be directly used as the prompt of the language model. Figure 2 shows an example of the visual clues. We observe that the information near the end of the prompt will have a more significant influence on the language model output. As the tags $\mathcal{T}$ are usually more informative and the local extractions are noisier, we input the visual clues with the order of local descriptions, caption, and tags.

To incorporate each local description, a naive way is to inject the coordinates of the bounding boxes directly into the prompt. However, we find the current language models still lack the capability to handle the inference task with numbers, especially in a zero-shot manner. Therefore, we reformat the bounding boxes $b_{\ell_k}$ into plain language by describing its location and size. Specifically, we adopt rule-based method to divide the locations into 9 classes {"*upper left*", "*upper middle*", "*upper right*", "*left*", "*middle*", "*right*", "*lower left*", "*lower middle*", "*lower right*"}, and divide the sizes into 3 classes {"*large*", "*moderate-sized*", "*small*"}, and incorporate these descriptions into the prompt.

The other visual clues are inputted straightforwardly in the format as showed Figure 2. The prompt is then fed into a large-scale language model to synthesize $K$ candidate paragraphs $\{T_i\}_{i=1}^K$ full of descriptive details.

### 3.3 Candidate Selection

Finally, we use the vision-language model again, to select the candidate that aligns best with the image,

$$S = \underset{T_i, i=1,\cdots,K}{\arg\max} \langle f_v(I), f_t(T_i) \rangle. \tag{5}$$

To further rule out the unrelated concepts in $S$, we filter the output again in sentence level. This is because large language models sometimes have hallucination issues, i.e., it might generate unrelated

sentences in the paragraphs. For example, a paragraph beginning with "*A couple is hugging on the beach.*" is likely to be followed with "*It's a beautiful day and they're enjoying the sun and each other's company.*" even if there is no visual clue suggesting the weather. Therefore, we split it into sentences $(s_1, s_2, \cdots, s_U)$, and use a threshold $\gamma$ to remove the sentences with lower similarities,

$$T^* = (s_i | \langle f_v(I), f_t(s_i) \rangle > \gamma, i = 1, \cdots, U). \qquad (6)$$

In this way, we obtain the final output $T^*$.

## 4 Automatic Evaluation Metric: SPIPE

As indicated by Figure 1, the generated paragraphs of images can be very flexible. This makes the n-gram based metrics, e.g., BLEU (Papineni et al., 2002), ROUGE (Lin, 2004), CIDEr (Vedantam et al., 2015), METEOR (Denkowski and Lavie, 2014), unsuitable for evaluating the generated text. Instead, we focus on the *semantic propositional content*. For example, given an image with content "*A man sitting in front of a blue snowboard*", a good evaluation metric for IPC should evaluate whether each of the semantic propositions is correct, namely, a). a man is sitting; b). a man is in front of a snowboard; c). the snowboard is

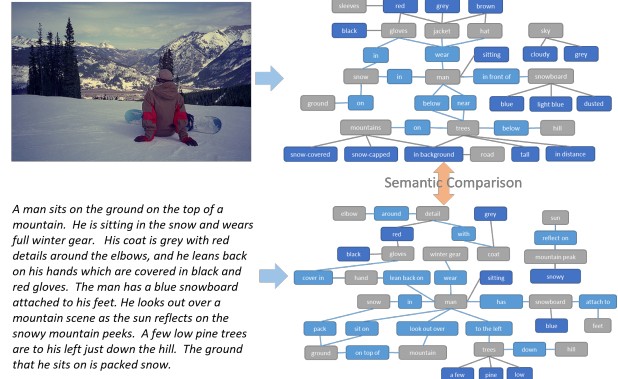

Figure 3: An example of the human-annotated graph and the text extracted graph.

blue, instead of the exact words used in the text. To do so, SPICE (Anderson et al., 2016) extracts the *scene graphs* (Johnson et al., 2015) from the generated texts and the reference texts, respectively, and computes an F-score between the graphs. SPICE targets image caption tasks, where there are usually multiple good references for each image, and the generation is less flexible. However, IPC tasks usually only have one reference (Krause et al., 2017), which is not enough to evaluate the flexible generation. Therefore, we propose to directly compare the scene graphs extracted from the generated text to human-annotated graphs. Figure 3 shows an example of the generated graph from text and the human-annotated graph for the image.

Specifically, a scene graph consists of the objects, the attributes of the objects, and the relationships between the objects. To parse the generated text into a scene graph, we use a two-stage approach following Anderson et al. (2016). First, we use the pretrained dependency parser (Klein and Manning, 2003) to establish the synthetic dependency between the words. Then we map from the dependency trees to scene graphs using a rule-based system (Schuster et al., 2015). Given scene graphs extracted from the text and the human-annotated graphs (Krishna et al., 2017), our metric computes an F-score based on the synonym match[2] Denkowski and Lavie (2014) between the two graphs among the conjunction of three sets of concepts: (object), (object, attribute), and (object, relationship, subject). Paying homage to Anderson et al. (2016), we name our approach **SPIPE**, Semantic Propositional Image Paragraph Evaluation.

## 5 Empirical Analysis

The basic evaluation of the generated output should include three aspects:

1. Accuracy. Most of the contents appearing in the paragraph should be from the image;
2. Completeness. Most of the contents appearing in the image should be included in the paragraph;
3. Coherence. Paragraphs should be more than concatenating the sentences together.

We evaluate the accuracy and completeness of the generated descriptions using the proposed automatic evaluation metric SPIPE, and do human evaluation to quantify the coherence. We include 500 randomly sampled outputs in `output.html` in the **Supplement Materials** for readers to perform a qualitative study.

---

[2]Tuples are considered to be matched if their lemmatized word forms are equal or if they are found in the same WordNet (Miller, 1995) synset.

Table 1: Comparison between different methods using SPIPE metric on the test set of the *Stanford dataset* (Krause et al., 2017).

|  | Name | F-score | Precision | Recall |
|---|---|---|---|---|
| Models | BLIP-large | 7.6 | 38.0 | 4.4 |
|  | Socratic model | 3.2 | 13.9 | 1.9 |
|  | BEST–general domain | 8.8 | 15.3 | 6.6 |
|  | BEST–VG domain | **10.0** | 17.5 | 7.6 |
| Oracle | BEST with human extracted visual clues | 22.9 | 32.8 | 19.0 |
| Annotation | Stanford dataset | 17.3 | 27.7 | 14.0 |
|  | Concatenation of VG captions | 18.9 | 40.0 | 14.1 |

## 5.1 Model Specification

**Models.** We adopt Florence-H (Yuan et al., 2021) as the vision-language model, BLIP-large (Li et al., 2022) finetuned on COCO captions dataset (Chen et al., 2015) as the captioner with its default setting, and one-stage detector as a general object detector. To be more specific about the detector, we first omit the category information from COCO (Chen et al., 2015) datatset and train Dynamic Head (Dai et al., 2021) on the bounding boxes only to formulate a class-agnostic object detector. We then use non-maximum suppression (NMS) to select the top 100 object proposals.

We use GPT-3 (Brown et al., 2020) *Davinci-text-001* model as the language model. To enable more difference in the generated candidates, we adopt temperature as $0.8$, as a higher temperature encourages the model to have more creative outputs. We adopt the frequency penalty as $0.5$ and the maximum number of tokens as $100$.

**Customized sets.** To construct a general domain tag set, we collect the most frequently searched 400 thousand queries in Bing Search as the tags $\{t_i\}_{i=1}^N$. We adopt the attribute set of the Visual Genome dataset (Krishna et al., 2017) as the attribute set $\{a_i\}_{i=1}^V$.

**Parameters.** We adopt number of tags $M = 5$, thresholds $\beta = \gamma = 0.2$, and number of candidates $K = 40$. Among $K = 40$ candidates, half of them are generated without caption information while the remaining half are with them. This is because we notice the caption model sometimes cannot output good captions due to too small bounding boxes. We also remove the bounding boxes that are smaller than $1/400$ of the image sizes.

## 5.2 Automatic Evaluation

In this section, we use SPIPE to benchmark the accuracy and completeness of our framework. We evaluate our framework on the test set of *Stanford dataset* (Krause et al., 2017). The dataset is a subset of Visual Genome (VG) dataset[3] (Krishna et al., 2017), and therefore we can obtain the human-annotated scene graphs from VG as well. We compare the following frameworks.

**BLIP** (Li et al., 2022). This is the BLIP-large model finetuned on COCO captions dataset.

**Socratic model** (Zeng et al., 2022). We adopt the image captioning code[4] without alternation.

**BEST-general domain**. This is our framework with the customized set listed above.

**BEST-VG domain**. With open-vocabulary capability, our framework can adapt to a specific domain. Here, we replace the customized tag set $\{t_i\}_{i=1}^N$ for the local objects as the object set of VG datasets.

The results are shown in Table 1. Our general domain framework significantly outperforms the BLIP model and the Socratic model. With the domain specified to VG, the performance is further boosted.

Figure 4 shows an example with a image cropped from the Socratic model paper (Zeng et al., 2022) directly. We find that caption generation does not require the complex prompt used in Socratic model. Our framework with only tagging information $\mathcal{T}$ can generate texts with a similar degree of detail. See Appendix D for more discussion.

---

[3]We remark that the VG caption data is included in the pretraining data of BLIP model. Therefore we do not claim our framework as a *zero-shot* method, despite that it can handle the images in the wild in a zero-shot way.

[4]`https://github.com/google-research/google-research/tree/master/socraticmodels`

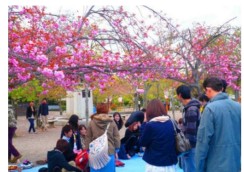

**BEST**: *This image features a group of people standing under a tree with pink blossoms. The people are all dressed in various types of clothing, and some are carrying bags or backpacks. The tree is a sakura tree, and the blossoms are in full bloom. There is a bench in the background, and the ground is covered in fallen petals.*
**SM**: *People gather under a blossoming cherry tree, enjoying the beauty of natural together.*
**BLIP**: *A group of people standing around a blue table.*
**BEST with only tags**: *The image is of people gathered in a park, looking at cherry blossoms. The parks are full of people enjoying the festival and the beautiful view.*

Figure 4: An example cropped from Zeng et al. (2022) paper, with other outputs for comparison.

To evaluate the representation capability of our visual clues, we also compare it to a naive scene graph generation method. We use the vision-language model to assign objects, attributes, and relationships between the objects, using the object set, attribute set, and relationship set of VG. And then we compare the generated scene graph to the human-annotated graph. The F-score is $0.3$, with precision $0.8$ and recall $0.2$. We discuss more on why this does not work in Appendix D.

We also build an oracle model to see the limit of our framework. The oracle model in Table 1 uses the ground truth objects with ground truth attributes to replace the corresponding concepts in the visual clues of our framework. It significantly outperforms the human annotation, either from Stanford dataset or from VG. This reveals the large potential of BEST with the development of object detectors.

## 5.3 Ablation

We perform an ablation study to see how each of the components contributes to the performance. Especially, we consider replacing the open-vocabulary object detector with YOLO v5 (Jocher, 2020), which is a closed-set detector trained with COCO classes. Table 2 shows the results. The performance of the YOLO v5 alternation is competitive compared to our general domain version. The precision is higher, which may be a consequence that YOLO models tend to recognize fewer objects (Zou et al., 2019). However, it is still inferior to our VG domain model.

Table 2: Ablation on each components. The metrics are F-score (F), Precision (P), and Recall (R).

| Name | F | P | R |
|---|---|---|---|
| BEST-VG domain | 10.0 | 17.5 | 7.6 |
| Extraction with YOLOv5 | 9.0 | 19.0 | 6.3 |
| Remove local information | 8.0 | 14.4 | 6.0 |
| Remove caption model $c(\cdot)$ | 8.7 | 15.0 | 6.6 |
| Input tags $\mathcal{T}$ only | 5.9 | 10.7 | 4.4 |
| Smaller language model (*curie*) | 8.9 | 15.9 | 6.7 |
| Weaker tagger (CLIP *ViT-L/14*) | 7.8 | 16.4 | 5.6 |

## 5.4 Human Evaluation

To further evaluate our framework, we perform human evaluation. We first compare BEST to human annotation. Specifically, we randomly sample 200 descriptions from the test set of the two sources. For each assignment, we present one image and two corresponding descriptions, and ask human evaluators to evaluate on accuracy, completeness, coherence, and ask an additional question "*which of the descriptions is written by human*" for the humanlikeness aspect. They will choose one answer from {*Description 1, Description 2, Cannot determine*}. For each assignment, we hire 5 workers using the Amazon Mechanical Turk platform. More details can be found in Appendix E. As the difference between the long texts can be subtle, we perform two statistical tests to see whether the difference is statistically significant. Please refer to Appendix E.2 for the hypotheses.

Table 3 shows the results. There is *no* significant evidence ($p$ value $\approx 0.5$) showing human annotation is better than BEST in terms of completeness and humanlikeness. However, BEST still falls behind in terms of accuracy and coherence. The failure cases are usually because the BEST outputs might contain small mistakes that cannot be easily filtered out, mostly from the hallucination of the language model. We show more examples in Appendix C.

We then compare BEST to BLIP and the Socratic model using similar hypothesis tests. The results show BEST are significantly better than BLIP and Socratic models under most of the metrics ($p$ value $< 0.05$). Note that here accuracy is defined slightly different than the precision used in Table 1: In human evaluation, providing background information about concepts in the image is not viewed as inaccurate, while in Table 1 it will hurt the precision.

Table 3: Human evaluation. $p$-value 1 is with the binomial test, and $p$-value 2 is with Mann–Whitney $U$ test. The blue regions in the voted proportion section represent the proportion that the descriptions from the first source are better than the second, while the orange ones represent the second are better than the first. The $1.2\%$ and $0.6\%$ in the middle of row 4 and 12 represent "*Cannot determine*".

| Sources | Criteria | Voted proportion | | | $p$-value 1 | $p$-value 2 |
|---|---|---|---|---|---|---|
| Anno. / BEST | Accuracy | 61.0% | | 39.0% | $2 \times 10^{-3}$ | $9 \times 10^{-6}$ |
| | Completeness | 51.5% | | 48.5% | 0.38 | 0.50 |
| | Coherence | 59.7% | | 40.3% | $5 \times 10^{-3}$ | $8 \times 10^{-4}$ |
| | Humanlikeness | 54.7% | 1.2% | 44.2% | 0.10 | 0.50 |
| BEST / BLIP | Accuracy | 57.2% | | 42.8% | 0.03 | $3 \times 10^{-3}$ |
| | Completeness | 73.6% | | 26.4% | $2 \times 10^{-11}$ | $2 \times 10^{-28}$ |
| | Coherence | 56.2% | | 43.8% | 0.05 | $5 \times 10^{-3}$ |
| | Humanlikeness | 58.3% | | 41.7% | $9 \times 10^{-3}$ | $5 \times 10^{-4}$ |
| BEST / Socratic | Accuracy | 59.4% | | 40.6% | $6 \times 10^{-3}$ | $7 \times 10^{-5}$ |
| | Completeness | 71.3% | | 28.7% | $2 \times 10^{-9}$ | $1 \times 10^{-24}$ |
| | Coherence | 68.7% | | 31.3% | $3 \times 10^{-7}$ | $2 \times 10^{-12}$ |
| | Humanlikeness | 53.4% | 0.6% | 46.0% | 0.18 | 0.35 |

## 6 Variants and Real-world Applications

The proposed framework opens up many creative real-world applications. For example, People with vision deficiencies may not be able to view images easily. BEST can help convert it into precise and comprehensive descriptions for general domain images.

Another example is the closed-loop training of the large models. The large-scale vision-language model and language model used in BEST are trained on tremendous amounts of data, and thus can memorize knowledge beyond human capability. We can use it to automatically annotate data, which is easy to scale up.

Table 4: Finetune BLIP-large on different data.

| Name | F | P | R |
|---|---|---|---|
| No finetune | 7.6 | 38.0 | 4.4 |
| With Socratic generated data | 3.9 | 17.5 | 2.3 |
| With BEST generated data | 11.6 | 23.1 | 8.2 |

Furthermore, it can incorporate commonsense knowledge into the text naturally. For example, in the second example of Figure 1, the text contains "*This image captures the everyday life of Cubans, with their traditional horse-drawn carts still in use.*" This is because our tags contain "*Cuba*" and "*buggy*", and the language model knows traditional horse-drawn carts are still in use in Cuba. We finetune a BLIP-large model on our BEST generated data. The training images are similar to Stanford dataset (Krause et al., 2017), which is around 15 thousands. After finetuning, the F-score improves more than $50\%$.

With small modifications, the proposed framework enables us to free human labor for even more applications. To list a few examples,

**Visual storytelling.** As shown in Figure 5 (a), the framework can generate charming stories based on the input image. To do so, we simply change the end of the prompt to be "*Tell me a creative story:*".

**Automatic ads generation**: As shown in Figure 5 (b), with the framework, the merchants only need to upload an image, and make small modifications to the generated advertisement as wanted. As there is usually one product in an image, we adopt the number of input tags $M = 1$. We also change the end of the prompt to be "*Write a product description to sell in eBay or Amazon marketplace to get lots of engagement:*".

**Social media post.** As shown in Figure 5 (c), the framework can be a social media bot, which may alleviate the workload of internet celebrities. We change the end to be "*Social media post:*".

**Background generation.** As shown in Figure 5 (d), the framework can also be used to provide some background knowledge. To do so, we change the end of the prompt to be "*Textbook text:*".

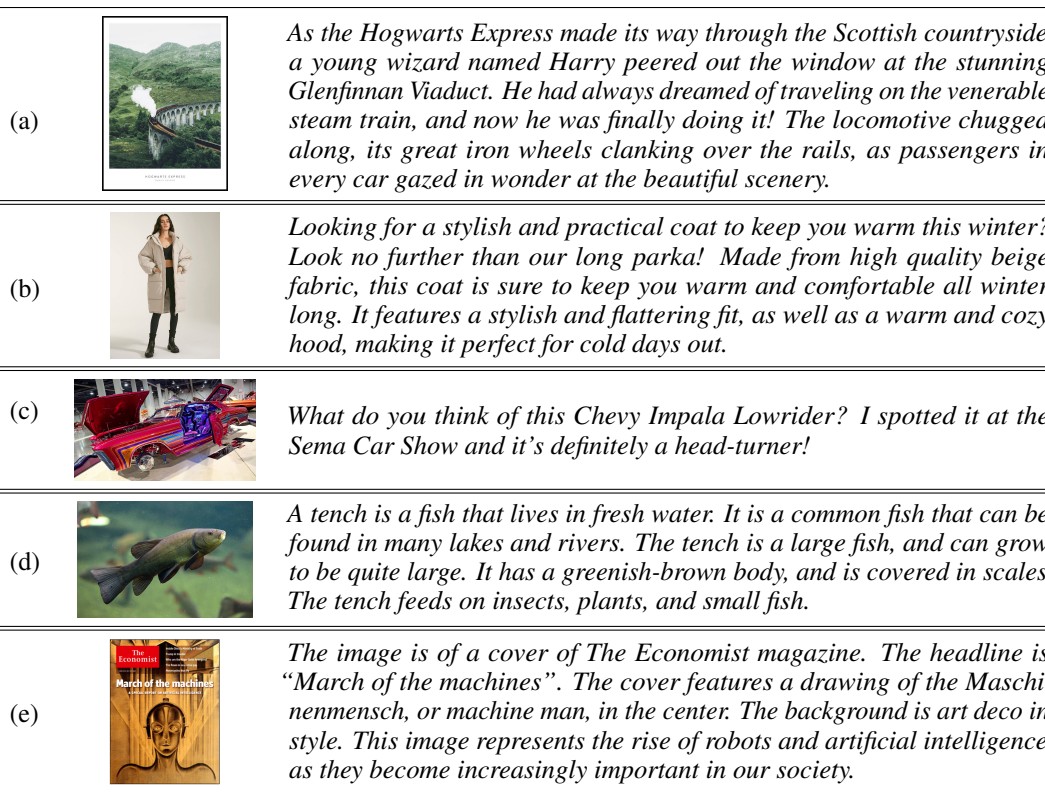

| (a) | | *As the Hogwarts Express made its way through the Scottish countryside, a young wizard named Harry peered out the window at the stunning Glenfinnan Viaduct. He had always dreamed of traveling on the venerable steam train, and now he was finally doing it! The locomotive chugged along, its great iron wheels clanking over the rails, as passengers in every car gazed in wonder at the beautiful scenery.* |
| (b) | | *Looking for a stylish and practical coat to keep you warm this winter? Look no further than our long parka! Made from high quality beige fabric, this coat is sure to keep you warm and comfortable all winter long. It features a stylish and flattering fit, as well as a warm and cozy hood, making it perfect for cold days out.* |
| (c) | | *What do you think of this Chevy Impala Lowrider? I spotted it at the Sema Car Show and it's definitely a head-turner!* |
| (d) | | *A tench is a fish that lives in fresh water. It is a common fish that can be found in many lakes and rivers. The tench is a large fish, and can grow to be quite large. It has a greenish-brown body, and is covered in scales. The tench feeds on insects, plants, and small fish.* |
| (e) | | *The image is of a cover of The Economist magazine. The headline is "March of the machines". The cover features a drawing of the Maschinenmensch, or machine man, in the center. The background is art deco in style. This image represents the rise of robots and artificial intelligence, as they become increasingly important in our society.* |

Figure 5: Examples for real-world applications with variants of the proposed framework.

**Applications with scene texts.** As shown in Figure 5 (e), when the input image has rich scene text, we plug in an OCR (Optical Character Recognition) model (Farley, 2022), and insert the OCR output into the prompts with prefix "*This image contains text:*" before the captions.

# 7 Applications to Visual Question Answering

The visual clues is a faithful and detailed description of the image, which can be used to answer visual questions leveraging the question answering ability of language models. Specifically, we replaced the ending of the prompt to be the question, *e.g.*, we replace the "*Describe the image in detail:*" by "*What is the man holding?*". We benchmark its performance in two Visual Question Answering (VQA) datasets – we use the GQA (Hudson and Manning, 2019) dataset for probing the capability of scene understanding, and the OK-VQA (Marino et al., 2019) dataset for the awareness of the commonsense knowledge.

Since no training is performed, BEST generated answer usually have different formatting from the ground truth, causing difficulty in evaluation. For example, for question "*Is the ground blue or brown?*", the ground truth answer in GQA is "*brown*", but the BEST answer is "*The ground in the image is brown.*". Therefore, we use GPT-3 model again to reformat the answer. We refer this evaluation method as *Generative*. Furthermore, for the GQA dataset, the answers in the training set and test set have a large overlap. So we adopt the nearest embedding from the training answers as the final answer, and refer the method as *Discriminative*. More details can be found in Appendix F.

Table 5 shows the evaluation results. BEST outperforms Socratic models significantly, suggesting our visual clues are better image representations. We also benchmark the accuracy on BLIP (finetuned on VQA v2 dataset (Goyal et al., 2017) and Visual Genome dataset (Krishna et al., 2017)) for reference, which is not directly comparable since its pretrain and finetune datasets have a significant overlap with the evaluation datasets. Figure 6 and Figure 7 show some success and failure cases of from the VQA datasets.

Table 5: The VQA accuracy on GQA and OK-VQA datasets.

| Method | Evaluation | GQA | OK-VQA |
|---|---|---|---|
| Socratic | Generative | 24.95 | 16.50 |
| | Discriminative | 26.89 | – |
| Visual Clues | Generative | 37.00 | **28.89** |
| | Discriminative | **39.93** | – |
| BLIP | Exact Match | 47.58 | 43.62 |

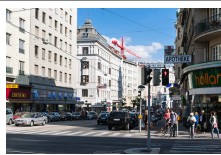 **Question:** *What material is the crosswalk in front of the stores?* **GT:** *Concrete* **BEST:** *Concrete*

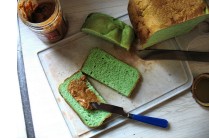 **Question:** *What ingredient is missing from the picture to make a pb and j sandwich?* **GT:** *Jelly* **BEST:** *Jelly*

Figure 6: Examples of the success cases from the GQA (left) and the OK-VQA (right).

# 8 Limitations and Further Improvements

**Prompt tuning.** As suggested in Brown et al. (2020), language models can infer better when they are shown examples in the prompt. In our experiments, however, this results in model directly copying sentence pieces from example paragraphs, introducing unnecessary noise. We suspect this prompt tuning approach may work better if the input examples are similar to the generated one. This may be a promising direction as we can better control writing style.

**Visual clues as a scene graph.** Our visual clue extraction process is motivated by the fact that an image can be comprehensively represented by a scene graph (Johnson et al., 2015). As mentioned in Section 4, a scene graph contains objects, attributes of objects, and the relationship between objects. In BEST, however, we do not include the relationships, as we observe in our initial study that the current vision-language models, although powerful, are not good at inferring relationships (echoing findings from Thrush et al. (2022)). Yet, relationships among the objects are important components of an image. This can be plugged into our framework if better vision-language models are developed.

**A well-trained filter model.** We find that the current filtering strategy (6) is not immune to certain types of mistakes. As also mentioned in Thrush et al. (2022), the vision-language model cannot accurately associate attributes to their corresponding objects. For example, in the second image of Figure 1, if there is a sentence "*The man wears a black shirt.*", it will lead to a high image-text relevance score, since there is a man, a shirt, and dark bush in the image. To handle this issue, we crop the image into local regions and pair each region with an attribute. Still, if it is the language model who hallucinates new attributes and the attributes happen to be in the image, these captions cannot be filtered out. We suspect an adversarially trained filter is needed to perform the task.

**Broader Societal Impact.** BEST inherits the risks of large vision and language models. BEST can potentially output offensive language and propagate social biases and stereotypes. For real applications, we can use rule-based methods or train a specific filter to filter out the offensive text. This is an area that we plan to explore to gain more insights further.

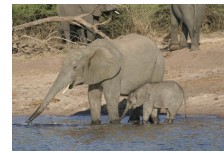 **Question:** *What place is pictured?* **GT:** *Shore* **BEST:** *Africa*

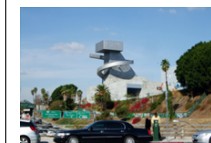 **Question:** *Where in the world would you find this structure?* **GT:** *California* **BEST:** *Los Angeles, California*

Figure 7: Examples of the failure cases from the GQA (left) and the OK-VQA (right).

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

# References

