# OpenReview forum: "Visual Clues: Bridging Vision and Language Foundations for Image Paragraph Captioning"
_NeurIPS.cc/2022/Conference — NeurIPS 2022 Accept_

### Official Review · Reviewer_WzAU · 2022-07-04

**Rating:** 6
**Confidence:** 3
**Soundness:** 3 good
**Presentation:** 3 good
**Contribution:** 3 good

**Summary:**

This paper introduces a new method for image paragraph captioning, where the goal is to generate long, coherent and informative textual descriptions of an image. Their method uses several large pretrained models, combined without the need for further training. From the image, open-vocabulary models like CLIP generate tags, object detectors generate a list of objects along with their locations, and image-captioning models generate (shorter) captions of the entire image or image regions. All of these outputs are combined into a natural language prompt (called "visual clues" by the authors), which is then fed to a language model like GPT-3. The language model then generates multiple candidate captions from the visual clues, which are then ranked by a contrastive model like CLIP. The authors also propose SPIPE, a new evaluation metric for measuring performance on image paragraph captioning. Using this metric, the authors show that their method outperform other baselines. The authors also conduct human evaluations, showing that their method typically outperforms previous ones, and can be competitive with human annotations.

**Questions:**

1) Is SPIPE correlated with human perception?

2) Have authors considered exploring tasks beyond IPC? For instance, VQA.

**Limitations:**

To the best of my knowledge, authors adequately discuss the limitations of their work.

**Strengths And Weaknesses:**

**Strengths**

1) The ideas presented in this work are (to the best of my knowledge) novel, and the research direction is timely.

2) The experimental setup and results are solid.

3) The paper is very clear and well written.

**Weaknesses**

1) One concern is that a great portion of the claims that their method is better than previous work rely on a metric proposed by this own work.  The authors explain why other evaluation metrics might not be ideal, but I believe it would still be informative to report them in the paper, especially since there exists image paragraph captioning benchmarks that use other evaluation metrics.

---

> ### Author Response · Authors · 2022-08-02
> **Response to Reviewer WzAU**
>
> Thanks for the constructive feedback!
>
> ### Q1: Other metrics
> Please refer to response to all reviewers: Other metrics besides SPIPE. We reported other metrics in Appendix B.
>
> ### Q2: Is SPIPE correlated with human perception?
> From our observation, yes. For example, SPIPE suggests that in terms of completeness, human annotations>BEST>BLIP>Socratic model. For human evaluation, we have human annotations>BEST, BEST>BLIP, BEST>Socratic model. The conclusion is in accordance. Unfortunately, we are not able to perform large-scale comparisons on correlations to human evaluation, as it requires at least dozens of IPC methods to construct a statistically meaningful comparison.
>
> ### Q3: Have authors considered exploring tasks beyond IPC? For instance, VQA.
> Yes. The visual clues are a faithful and detailed description of the image, which can be used to answer visual questions leveraging the question answering ability of language models. Specifically, we replaced the ending of the prompt to be the question, e.g., we replace the “ *Describe the image in detail:* ” by “ *What is the man holding?* ”. We benchmark its performance in two Visual Question Answering (VQA) datasets -- we use the GQA dataset for probing the capability of scene understanding, and the OK-VQA dataset for the awareness of the commonsense knowledge.
>
> The table below shows the evaluation results. BEST outperforms Socratic models significantly, suggesting our visual clues are better image representations. We also benchmark the accuracy on BLIP (finetuned on VQA v2 dataset and Visual Genome dataset) for reference, which is not directly comparable since its pretrain and finetune datasets have a significant overlap with the evaluation datasets.
>
> |     Dataset    |     Method                  |     Evaluation        |     Accuracy    |
> |----------------|-----------------------------|-----------------------|-----------------|
> |     GQA        |     Socratic model          |     Generative        |     24.95       |
> |                |                             |     Discriminative    |     26.89       |
> |                |     BEST                    |     Generative        |     37.00       |
> |                |                             |     Discriminative    |     39.93       |
> |                |     BLIP (not zero-shot)    |     Exact match       |     47.58       |
> |     OKVQA      |     Socratic model          |     Generative        |     16.50       |
> |                |     BEST                    |     Generative        |     28.89       |
> |                |     BLIP                    |     Exact match       |     43.62       |
>
> The experiment details can be found in Appendix F. We observe that some of the failure cases are not actually wrong answers. For example, for an image with elephant drinking water by a river, and question “ *What place is pictured?* ”, the ground truth answer is “ *Shore* ”, while our generated answer is “ *Africa* ”, which is also not wrong.

---

> ### Comment · Reviewer_WzAU · 2022-08-08
> **Thank you for the response**
>
> Thank you for your time to think about my comments and suggestions. I stick with my pre-rebuttal rating as I have already recommended acceptance.

---

> > ### Author Response · Authors · 2022-08-08
> > **Thanks for your comments and suggestion!**
> >
> > Thanks a lot for your comments and suggestion! We appreciate your contribution in improving our work!

---

### Official Review · Reviewer_8CQ9 · 2022-07-11

**Rating:** 6
**Confidence:** 4
**Soundness:** 3 good
**Presentation:** 3 good
**Contribution:** 3 good

**Summary:**

The authors propose a new framework for generating image paragraph captioning in this paper. The framework extracts text descriptions from images using different pretraining modules, subsequently feeds these extracted modules to a language model for paragraph generation, and finally selects the optimal paragraph from several candidate results using a cross-modal selection module as the final result. Also, to better measure this task's results, the authors propose a new evaluation method SPIPE based on the SPICE evaluation method.

**Questions:**

See the “weakness”.

Hope the authors improve this paper according to the “weakness”.

**Limitations:**

The authors have addressed some of the limitations and potential negative societal impact of their work.

**Strengths And Weaknesses:**

Strengths
1. The framework proposed in this paper can generate a paragraph for a given image, which is much longer than the previously widely used image caption.
2. The authors propose a new evaluation metric SPIPE to measure the results of IPC.

Weakness
1. Lack of comparison with other traditional IPC methods. There are many previous IPC methods mentioned in the Related Works section.   but their results are not compared with the proposed methods' results in the experimental section.
2. I think this work seems like a direct extension of PICa to the IPC task. The novelty may be limited.
3. There is no experimental proof in the article that the new evaluation metric SPIPE could evaluate IPC better than the previous metrics, such as SPICE and METEOR. There is also no analysis of whether the SPIPE evaluation metrics can evaluate different aspects. Besides, SPIPE needs human-annotated graphs for each paragraph in the dataset, which limits the generalizability of this metric. As a result, only using the SPIPE metric in all the experiments is not convincing.
4. In Table1, compared with BLIP-large, BEST–general domain improve 1.2 in F-score. Due to more processes (Candidate Synthesis&Selection) and higher demands on computing resources (GPT-3), I don't think the improvement is significant.

---

> ### Author Response · Authors · 2022-08-02
> **Response to Reviewer 8CQ9 -- Part 1**
>
> Thanks for the valuable comments!
>
> We would like to first highlight that our framework **does not involve any training or training data**, which is essentially different from other IPC methods and PICa.
>
> ### Q1: Comparison with traditional IPC methods
> The settings of BEST and traditional IPC methods are unfair to compare, since traditional IPC methods are fully trained on labeled task data while our approach is almost zero-shot. As a result, the current metrics on these benchmarks are hard to provide a fair score to two language styles although their meanings are the same to human justification. For example, human annotators without domain specific knowledge tend to say, “ *The man is riding a white surfboard* ”, while our framework outputs “ *This image captures a surfer performing a cutback maneuver* ”. For the current metrics, the difference in language styles would lower the scores of our framework compared with IPC since their well-trained language style is closer to the human annotation. As a matter of fact, our BEST scores in Appendix B Table 5 are not as good as IPC methods. For example, the scores for paper [r1] are BLEU-4 10.58, METEOR 17.86, CIDEr 30.63.
>
> As a matter of fact, our framework and traditional IPC methods are essentially different pipelines. Our framework has unique advantages that traditional IPC methods do not have:
>
> 1. **Easier implementation**, as discussed in response to all reviewers computation cost.
> 2. **Versatile to different application scenarios**, as suggested by Figure 5.
> 3. **Robust to domain shift.** As suggested Figure 1 of [r1], for a black and white image which the model is not trained on, the inference of the well-trained model does not make sense. The output is like:
> > *A man is skateboarding on a skateboard. He is wearing a black shirt and black pants. He is wearing a black cap and a black hat. A man is wearing a black cap and a black shirt. A man is wearing a black shirt and a black pants. A man is wearing a black shirt and a black pants. A man is wearing a black shirt and a black pants…… A man is wearing a black shirt and a black pants.*
>
> &ensp;&ensp;&ensp; Yet the man actually wears a white hat. This is a common issue of models trained on small language datasets.
> In comparison, our Figure 9 and 10 include some black and white examples.
>
> 4. **Better language quality.** The large language models are trained on tremendous amount of data, leading to much better capability in text generation. Figure 6, 8, 9, and 10 show a few examples of the smooth, coherent paragraphs. In contrast, here is a successful example from Figure 1 of [r1]:
> > *Two people are sitting on a bench. The elephant is sitting on the dirt. The man is sitting on top of the elephant. The woman is wearing a white shirt. The man is wearing a black shirt. There is a tree behind the elephant. There are trees on the ground. There are trees in the background.*
>
> &ensp;&ensp;&ensp; We cannot even tell how many people are there according to the text (as a matter of fact, there are two.), due to lack of coherence among the sentences.
>
> 5. **Awareness of commonsense knowledge.** As suggested by Figure 1, our framework output background knowledge like “ *This image captures the everyday life of Cubans, with their traditional horse-drawn carts still in use.* ”, which smoothly complete the paragraph. Traditional IPC methods cannot have enough training data to enable such capability.
>
> We have added this discussion in Appendix D to prevent confusion for the readers.
>
>  [r1] Luke Melas-Kyriazi, Alexander Rush, George Han. 2018. Training for Diversity in Image Paragraph Captioning. (Although written in 2018, this work rank #2 in Paper With Code: https://paperswithcode.com/sota/image-paragraph-captioning-on-image-paragraph)

---

> > ### Author Response · Authors · 2022-08-02
> > **Response to Reviewer 8CQ9 -- Part 2**
> >
> > ### Q2: Difference from PICa
> > We list the differences between PICa and BEST as follows:
> >
> > 1. To answer the visual questions, we only need to extract the key visual concepts from the image, while IPC requires a comprehensive understanding of the image. It is relatively easy to represent the key concepts using text prompt, while it is not trivial to exhaust the details in an image -- after all, an image is worth a thousand words.
> > 2. As a VQA method, PICa outputs deterministic single words or simple phrases, while our framework outputs creative long texts. This introduces many difficulties – how to control the level of detailedness? How to remove hallucination while maintaining creativeness? How to define a “good” output?
> > 3. PICa exploits the whole training set of OK-VQA to construct the prompt examples, while we do not involve any training data.
> >
> > The only common ground between BEST and PICa is to use GPT-3 to obtain the final outputs. Our work is NOT a direct extension of PICa.
> >
> > ### Q3: On the metric SPIPE
> > “ *There is no experimental proof in the article that the new evaluation metric SPIPE could evaluate IPC better than the previous metrics, such as SPICE and METEOR.* ”
> >
> > Please refer to response to all reviewers: Other metrics besides SPIPE. We have added discussions on why other metrics cannot work well here, and also reported other metrics.
> >
> > As shown in the SPICE paper [r2], evaluation metrics based on semantic propositional contents can achieve better correlation with human evaluation. SPICE takes reference text for comparison, because for common captioning benchmark, there are multiple high-quality references, which together form a good description to the images. In IPC task, however, the reference is far from enough. In contrast, the scene graphs form a better description for the image. That is the reason we extend the SPICE metric to our SPIPE metric.
> >
> > We understand that we do not perform large-scale comparisons on correlations to human evaluation, like what SPICE did. Not only because the conclusion is already verified in the SPICE paper, but also because it requires at least dozens of IPC methods to construct a statistically meaningful comparison, which is not available online. Although not statistically meaningful, we can clearly observe some of the correlations between SPIPE and human evaluation: SPIPE suggests that in terms of completeness, human annotations>BEST>BLIP>Socratic model. For human evaluation, we have human annotations>BEST, BEST>BLIP, BEST>Socratic model. The conclusion is in accordance.
> >
> > “ *There is also no analysis of whether the SPIPE evaluation metrics can evaluate different aspects.* ”
> >
> > Precision is defined as the fraction of visual concepts related to the image among the concepts in the generated paragraph, and recall is the fraction of visual concepts in the paragraph among all the image concepts. It is intuitive that precision and recall corresponds to accuracy and completeness.
> >
> > “ *Besides, SPIPE needs human-annotated graphs for each paragraph in the dataset, which limits the generalizability of this metric.* ”
> >
> > Other existing evaluation methods require human-annotated texts. To the best of our knowledge, all commonly adopted automatic evaluation metrics require human annotations.
> >
> > “ *As a result, only using the SPIPE metric in all the experiments is not convincing.* ”
> >
> > The golden standard of text evaluation is not any of the evaluation metrics, but human evaluations, and BEST output is much better than other baselines according to the human evaluations. It is not true we “ *only using the SPIPE metric* ”. In addition, we reported other metrics in Appendix B.
> >
> > [r2] Peter Anderson, Basura Fernando, Mark Johnson, Stephen Gould. 2016. SPICE: Semantic Propositional Image Caption Evaluation.

---

> > > ### Author Response · Authors · 2022-08-02
> > > **Response to Reviewer 8CQ9 -- Part 3**
> > >
> > > ### Q3: Improvement over BLIP is not significant
> > > Please refer to response to all reviewers for our discussion on the computation cost. Again, although GPT-3 seems to be a large model, with the development of cloud services, all practitioners need to do is to call the APIs.
> > >
> > > We disagree that our improvement over BLIP is not significant. Quantitatively, BLIP is trained on VG data, so is well-aligned with the domain. In contrast, BEST-general domain is not leaning towards any domains and is capable of handling images in the wild. Still, the SPIPE F-score of BEST-general domain is 15.8% higher than BLIP.
> > >
> > > More importantly, human evaluation shows BEST is **statistically significantly** better than BLIP in every aspect: accuracy, completeness, coherence, and humanlikeness.
> > >
> > > Qualitatively, as shown in the examples in Figure 6, 8, 9, and 10, the level of detailedness and language quality of BLIP is not comparable to BEST. Take one example from Figure 6,
> > > > BEST: This image captures a surfer performing a cutback maneuver at the Superbank surf competition. The surfer is positioned in the middle of the frame and is relatively small in comparison to the surrounding waves. The waves are large and crashing, providing an impressive backdrop for the surfing action.
> > >
> > > > BLIP: A man riding a wave on top of a surfboard.
> > >
> > > BEST describes the scene vividly, while BLIP only provides a high-level summarization.

---

> ### Author Response · Authors · 2022-08-08
> **Is there any remaining concerns or questions we can address?**
>
> Thanks again for the great review!
>
> As the end of the rebuttal phase is approaching, we would like to double-check whether you have any remaining concerns or questions we can address. We are also happy to provide additional information or clarification if needed.
>
> Please let us know if you have any further questions or concerns :)

---

### Official Review · Reviewer_ErrM · 2022-07-11

**Rating:** 6
**Confidence:** 3
**Soundness:** 3 good
**Presentation:** 3 good
**Contribution:** 3 good

**Summary:**

The paper presents a general framework (BEST) for generating semantic visual clues by capturing information through tags, captions, object detections by leveraging existing pre-trained visual LMs and textual LMs. The framework enters them in an ascending order of their quality (highest quality predictions at the end): local descriptors, captions, and tags as input to a large-scale LM to generate K candidate paragraphs. The last step leverages the vision language model to select the top K candidates that align best with the image.  The paper presents an extensive set of experiments against state-of-the-art to show the effectiveness of the proposed framework. The paper has both ablation and qualitative results to demonstrate how each component contributes to the final performance and quality of the generated paragraphs. The paper also proposed an evaluation metric: SPIPE that captures Accuracy, Completeness and Coherence.


**Questions:**

Detailed questions/comments

In Figure 2, could authors highlight what color is being used to represent the text encoder as f_t(.)?

Line 35: Florence reference is missing. Please refer at the first introduction. There is a reference later e.g. on Line 106.

Line 112: How are tags generated {t_i}_{i=1}^{N}?
Section 3.1 add/create sub-subsections or bold headings to improve organization.

Line 147/148: Could more intuition be added to what sentences are kept or removed from Candidate selection Step?

Line 204: Some details on how tau temperature needs to be adopted in practice (intuition is needed)? What happens if \tau is 100 vs 0.8 as used in the paper?

Line 206: Tags are discussed. How you refer this back on Line 113 where it was unclear how these N tags are selected.

Some more clarification what are the exact inputs to BEST will improve reproduction of the results. What models need to be re-trained/re-run etc? As the paper is leveraging most  of the pre-trained models. It will be good to summarize in a Table/Figure how each component is participating in the final BEST model.


**Limitations:**

Already captured in Section 7 of the paper. Authors could  add some "additional societal biases that textual language models suffer from". Add some details on pros/cons of training these large models to environment.

**Strengths And Weaknesses:**

Strengths:
1. Overall, the paper is well written and is easy to read.
2. The paper presents a general framework (BEST) for generating detailed image caption paragraphs. The quantitive/qualitative evaluation results show the effective of BEST against BLIP-large and Socratic model in Table 1.
3.  Ablation results in Table 2 show the effectiveness of each component of BEST and Table 3 captures Accuracy, Completeness and Coherence components of BEST against Ground-truth/BLIP/Socratic.
4. Fine Tuning BLIP-large also improves results as shown in Table 4.
5. The paper presents many real-world setting where BEST could be useful. May detailed results are presented in Figure 5.
6. The Section 7 clearly calls out limitations and further improvements to the BEST.
Weakness:
1. Given Socrate model is solving a similar problem: it will be good to discuss how and why BEST is so much better than Socrates model in evaluation results.
2. In related work, more details around how BEST model is different than Socrate model with be different.
3. Given, there are many pre-trained model involved and also fine tuning is presented. How much it costs in terms of time/resources to train BEST? How much more expensive is training of BEST model when compared to Socrate model?

---

> ### Author Response · Authors · 2022-08-02
> **Response to Reviewer ErrM**
>
> Thanks for the valuable feedback!
>
> ### Q1: Why BEST is better than Socratic model
>
> The major issue of Socratic model is that its prompt contains inaccurate/useless information, and is not informative enough. Its prompt is
>
> *I am an intelligent image captioning bot. This image is a {img_type}. There {num_people}. I think this photo was taken at a {place1}, {place2}, or {place3}. I think there might be a {object1}, {object2}, {object3},... in this {img_type}. A creative caption I can generate to describe this image is:*
>
> There are three issues with this prompt:
>
> 1. Useless information. We find a much shorter prompt, e.g.,
> *"This is a {img_type} taken at {place1} containing {object1}, {object2}, {object3}… Generate a caption:"*
> has comparable (or even better) performance as the one above.
> 2. Inaccurate information. As we discussed in Section 7, VL models trained with contrastive loss cannot differentiate some details in the sentence. We observe that the *num_people* obtained by Socratic model is basically a random guess.
> 3. Lacking information. The *img_type* only contains four categories, *photo*, *cartoon*, *sketch*, *painting*. The object list is a list of common objects, not as comprehensive as ours. The place information usually redundant given the object information.
>
> In contrast, our prompt is much more informative and well-constructed. We have included this discussion in Appendix D.
>
> ### Q2: Training cost
> Our framework does **not** require training. We exploit the existing pretrained models directly for inference. Please refer to the response to all reviewers for more discussions on the computation cost.
>
> ### Q3: Color for $f_t(\cdot)$
> As introduced in Section 3.1, the open vocab tagger, i.e., the orange block in Figure 2, is composed of an image encoder$ f_v(\cdot)$ and a text encoder $f_t(\cdot)$. We have improved the descriptions in Section 3.1 to make it clearer.
>
> ### Q4: Missing Florence reference
> Thanks for pointing it out. We have added the reference in Line 35.
>
> ### Q5: How are tags generated
> As mentioned in Section 5.1, we collect the most frequently searched 400K queries in Bing Search as the input tag list.
>
> ### Q6: Add bold headings in Section 3.1
> Thanks for the great suggestion. We have added bold headings in Section 3.1.
>
> ### Q7: Intuition behind sentence removal when selecting candidate (Line 147/148)
> Large language models sometimes have hallucination issues, i.e., it might generate unrelated sentences in the paragraphs. For example, a paragraph beginning with " *A couple is hugging on the beach.*" is likely to be followed with " *It's a beautiful day and they're enjoying the sun and each other's company.*" even if there is no visual clue suggesting the weather. Therefore, we divide the generated paragraph into sentences, and process the sentences with the open vocab tagger again – if there is a sentence that is not aligned with the image, i.e., the alignment score between the sentence and the image is below $\gamma$, we remove it. We have added more explanation in Section 3.3 to make it clearer.
>
> ### Q8: Selection on $\tau$
> $\tau$ is the sampling temperature for GPT-3 model. GPT-3 allows a temperature from 0 to 1. Intuitively, a larger $\tau$ value encourages the model to have more creative outputs, while $\tau=0$ corresponds to argmax sampling, and is more suitable for well-defined answers (e.g., answers for " *What is one plus one?* "). Our application is creative in nature. Moreover, larger $\tau$ value promotes the generated multiple candidates to be more different, so we have a higher chance to select a good one. Therefore, we adopt $\tau=0.8$, which is a relatively high temperature. We have added more discussion in Section 5.1.
>
> ### Q9: Tag discussion on Line 206
> We believe it is better to introduce how the tags are collected in the experiment section. We want our framework to be general and can handle tag lists from different domains. For example, if the images are from medical domain, we can replace the tags with medical terminologies. Therefore, in section 3.1, we do not introduce how the tags are collected.
>
> ### Q10: Exact inputs to BEST
> Thanks for the great suggestion! We include a few prompt examples in Figure 7. It should be straightforward to construct the prompts by looking at the examples. We believe it would be easier to understand than plain tables. Again, there is no module re-trained. All we did was the inference.

---

> ### Author Response · Authors · 2022-08-08
> **Is there any remaining concerns or questions we can address?**
>
> Thanks again for the great review!
>
> As the end of the rebuttal phase is approaching, we would like to double-check whether you have any remaining concerns or questions we can address. We are also happy to provide additional information or clarification if needed.
>
> Please let us know if you have any further questions or concerns :)

---

### Official Review · Reviewer_b7uK · 2022-07-12

**Rating:** 6
**Confidence:** 5
**Soundness:** 4 excellent
**Presentation:** 4 excellent
**Contribution:** 3 good

**Summary:**

The visual clues paper tries to leverage the recent advances in both vision-and-language and natural language understand fields to generate comprehensive and rich descriptions of the images. The model works in three steps: (i) generate a set of visual clues (i.e. information) about the image to embed or encode rich visual information (ii) pass these visual clues as a prompt to a large language model to generate candidate paragraphs (iii) select the best paragraph from the candidates. The paper also purposes a new metric to better evaluate long descriptions using scene graph comparisons. The human and automatic evaluations both suggest that the descriptions generated by this method are almost on par with the human annotations.

**Questions:**

- It would be great to see the exact visual clues as well for the images in the supplementary. Without that context, it is hard to understand what was passed to GPT3 to generate the description.
- Will any information about the 400K top bing queries be provided?
- There is no basis behind claiming that n-gram based metrics don't work for larger paragraph generations for images.
- When the tags are matched with the images, are the tags directly matched or is some sort of prompt used?
- What is source/intuition behind the paragraph on line 38
- I disagree with footnote 2 and believe that this overlap is somewhat impactful. The Visual Genome captions are not just semantic propositions but are full captions. Also, in a single image, there are multiple crops for which we have the captions. Now, when these crops along with their captions compound, they do provide a lot of information about the image which is non-trivial leak
- On line 292, how much BEST-generated data was the BLIP-large model trained on?

**Limitations:**

Yes, the authors have discussed the limitations. In general, this model because it is developed on proprietary models will not be available so can't have direct impact.

**Strengths And Weaknesses:**

Legend: S (Strength), W (Weakness), C (General comment)
- C: The paper is very well written and easy to understand. Though, there are some things which are not clear and not well-mentioned probably due to the fact that some of the models used in this paper are proprietary.
- S: The paper connects the recent advances in the vision-and-language and NLP fields really well by leveraging almost only the pretrained models to generate very coherent and comprehensive descriptions of the images. Since there is almost no training involved, it further suggests how unimodally strong models can be bridged across modalities leveraging abundantly available unimodal data in the world.
- S: The approach to generate visual clues is quite exhaustive and covers various sources of visual information: tags using a contrastively trained VL model, object detection model whose bounding box classes are again calculated by the contrastively-trained VL model, and a full-image caption generator.
- S: One important contribution of this paper is a new automated metric for evaluating longer paragraph descriptions of the images as existing n-gram based metrics are not suitable for the variety seen in longer descriptions. The metric uses scene graph based similarity and is an extension to SPICE metric.
- S: Human evaluations on the outputs of the model suggest that model's outputs are almost on par in terms of completeness and humanlike but fall behind in accuracy and coherence.
- W: The previous point brings me to one of the weaknesses of this approach is that since we are relying on pretrained models in the complete pipeline, any errors are eventually compounding as we go across the pipeline which means that if any wrong content is passed via visual clues to the text generator, it would add a compound on top which is why maybe the accuracy and coherence are low.
- W: It is hard to understand the significance of SPIPE without seeing results on other metrics for the baselines as well. As of now, only SPIPE results are reported and there is no discussion on why ngram metrics probably don't work on the paragraph comparison.
- W: The model seems to ignore scene text in the image which might be an issue in real world scenarios as scene text is omnipresent. Even when you are reading this review, you are reading scene text.
- W: There have been works in past such as VisualGPT [1] which try to adapt the language model directly to image captioning. A comparison or discussion with them would be helpful.
- W: The baselines except BLIP are somewhat weak. Even BLIP when finetuned on COCO captions would be more inclined to generate shorter captions. This is also evident from the examples in supplementary. Since COCO captions are small, it would make more sense to use and compare against captioners trained on dataset which contain larger image descriptions such WIT and Localized Narratives which the paper doesn't do as of now. Most of the models used as baselines are tuned for generating single sentence captions.
- W: Most of the models used in this paper are proprietary and not available to the public so it is hard to make any comparison down the line. What would be interesting is to use CLIP instead of Florence along with a publicly available large language model to see how this approach performs.

[1] Chen, Jun, et al. "Visualgpt: Data-efficient adaptation of pretrained language models for image captioning." Proceedings of the IEEE/CVF Conference on Computer Vision and Pattern Recognition. 2022.

---

> ### Author Response · Authors · 2022-08-02
> **Response to Reviewer b7uK**
>
> Thanks for the constructive feedback! We appreciate your insight on bridging across modalities using unimodal models is a good way to exploit the data in the world.
>
> ### Q1: Errors are compounding
> Indeed, we observe that the errors in the final output are majorly due to the errors in local tagging / captioning module, since the local regions may be small, and therefore have a domain shift from their training data. To alleviate the issue, we filter out the too-small regions.
>
> Yet this actually suggests our framework has better interpretability. Consider an end-to-end encoder-decoder model, the improper configuration in encoder may also propagate to the final output. But if there are errors in the output, it is very difficult to diagnose whether the problem is in encoder or decoder. In contrast, visual clues explicitly show which module needs improvement. And thanks to our framework’s composability, if better local tagging and captioning modules appear in the future, we can easily replace the modules without any extra training cost.
>
> ### Q2: Other metrics
> Please see the response to all reviewers.
>
> ### Q3: Include scene text
> Thanks for the suggestion. We added one example in Section 6, where an OCR module is plugged in. For text-heavy scenarios, the variant with OCR indeed works better.
>
> ### Q4: VisualGPT
> We have added discussions in Section 2.
>
> ### Q5: Baseline for long captioner
> As you may notice, this is not a fair comparison, as our framework does not require image / long text pairs for training at all. Yet we agree that having a model trained on WIT or Localized Narratives for reference is useful. Do you mind sharing with us the model in your mind that should be used here for comparison? We will include the comparison in the next version.
>
> ### Q6: Proprietary components
> Two major proprietary components are the Florence model and the 400K bing queries. Both will be available to the public soon via API. Once they are publicly accessible, we will open-source our code as well.
>
> ### Q7: Non-proprietary pipelines
>
> GPT3 is publicly available via API. We replaced the Florence model with CLIP (Vi), and here are the scores:
>
> |     Tagger    |     F-score                  |    Precision        |     Recall    |
> |----------------|------------------|-----------------------|-----------------|
> |     Florence        |  10.0   |  17.5   |    7.6        |
> |         CLIP            |   7.8     |     16.4    |     5.6        |
>
> Qualitatively, the difference between Florence and CLIP lies in the accuracy of tagging. We observe there are more irrelevant tags in CLIP results than Florence.
>
> Also, we have tested our framework on fully open-sourced language models, e.g., GPT2 (gpt2-xl). But the output paragraphs do not make sense at all. As discussed in [r1], some capabilities of language models only emerge when the models are large enough. So GPT2 does not have the zero-shot visual clue understanding and summarization ability we use here.  However, GPT2 may still work with some finetuning.
>
> ### Q8: Examples for visual clues
> We have added a few examples of visual clues in Appendix A Figure 7.
>
> ### Q9: Prompt for tag match
> Yes, we followed the settings of CLIP.
>
> ### Q10: Intuition behind the paragraph on line 38
> “ *The visual clues are interpretable, not only for humans, but also for machines…… while not cluttered with irrelevant information from the visual clues.* ”
>
> This is based on our observation that large language models can process almost any textual information. For example, it generates reasonable output with inputs like “ *tell me more about Labrador* ” or “ *what does Mount Rainier look like* ”. So we believe that the model can digest visual clues and provide related information about the visual concepts.
>
> “ *Whereas this open-loop process could potentially suffer from object hallucination issues …… back to the original image.* ”
>
> The hallucination issues of language models are documented extensively in the literature, e.g., [r2][r3]. We have added the references to our paper.
>
> ### Q11: Data leakage
> We have removed the claims on this is not a data leakage concern. Indeed, this is the reason that we do not claim our framework as “ *zero-shot* ”, despite that it can handle the images in the wild in a zero-shot way. Yet it is too clumsy to retrain BLIP-large model without VG data.
>
> ### Q12: Synthetic data
> We generate 15K of synthetic data since this is just a proof-of-concept experiment. A larger amount of data may lead to better performance. We have added more experiment details in Section 6.
>
> [r1] Jason Wei, Yi Tay, Rishi Bommasani, et al. 2022. Emergent abilities of large language models
>
> [r2] Joshua Maynez, Shashi Narayan, Bernd Bohnet, and Ryan Thomas Mcdonald. 2020. On faithfulness and factuality in abstractive summarization.
>
> [r3] Chunting Zhou, Graham Neubig, Jiatao Gu, et al. 2020. Detecting Hallucinated Content in Conditional Neural Sequence Generation.

---

> > ### Comment · Reviewer_b7uK · 2022-08-09
> > **Thanks for the rebuttal**
> >
> > I would like to thank the authors for the rebuttal. About the baseline captioners, you can maybe take baselines in original BLIP paper such as LEMON or even some older models such as AoANet, but given the time frame, it is not needed. Most of my concerns are resolved and I still recommend acceptance for this paper.

---

> ### Author Response · Authors · 2022-08-08
> **Is there any remaining concerns or questions we can address?**
>
> Thanks again for the great review!
>
> As the end of the rebuttal phase is approaching, we would like to double-check whether you have any remaining concerns or questions we can address. We are also happy to provide additional information or clarification if needed.
>
> Please let us know if you have any further questions or concerns :)

---

### Author Response · Authors · 2022-08-02
**Response to All Reviewers**

We thank all the reviewers for their time and effort in reading and reviewing our paper.
Please kindly refer to the individual responses below for our response to each question. We have also updated our draft to reflect reviewers’ opinions in detail.

&nbsp;
### Other metrics besides SPIPE (To b7uK, 8CQ9 and wZAU)

**The other metrics were already reported in Appendix B (Table 5)**. The proposed framework still performs the best. From the table we can also see why the n-gram based methods do not work well – although BLIP and Socratic models have meaningful outputs, the resulted scores can be nearly zero. This is because, on the one hand, the output paragraphs can be very versatile. It is not meaningful to ask the models to output specific sentences/n-grams. On the other hand, the reference text is not of high quality. For each image, only one reference paragraph is provided (unlike other caption benchmarks, e.g., COCO). And there are usually many details missing in the reference text, not as exhaustive as the scene graphs.  Take the first image of Figure 6 as an example. Although the human annotated paragraph is already a long one, it does not mention the color of the table and chair, and there is a water bottle in the side of the backpack.

We also conducted human evaluation, which we believe should be the golden standard, instead of any automatic evaluation methods.

&nbsp;
### Computation cost (To ErrM and 8CQ9)

Our inference-only framework involves **no training** and **no data** for training. Therefore, the cost of building such a pipeline is much lower than traditional frameworks.

Traditionally to build a machine learning application, researchers need to 1). Collect a dataset; 2). Select a training framework; 3). Train the model with repeated hyper-parameter tuning. And then we can have a model that is specifically designed for such an application. If the business need is adjusted, the process needs to be repeated to accommodate the shifted application domain.

In contrast, what we propose is a light-weighted solution: The pretrained models are either available via API service (e.g., GPT3, captioner), or will be available soon (e.g., Florence tagger). To build our framework, researchers only need to plug in the APIs.  As suggested by Figure 5, it can handle various scenarios with only minor modification.

One may argue that some applications require fast inference, which our large pretrained model cannot handle. Yet, using the synthetic data generated by BEST (~20K), we have successfully replaced GPT-3 by a DeBERTa-large model with comparable performance. Using as a data generation pipeline, BEST is faster and more stable than recruiting human laborers.

---

### Author Response · Authors · 2022-08-02
**Summary of Changes**

Here is a summary of changes in our draft to reflect reviewers’ opinions in detail.

 - (ErrM) Section 1 line 35: Add Florence citation.
 - (b7uK) Section 1 paragraph 38: Adjust wording and add citation for the hallucination issue of language models.
 - (b7uK) Section 2: Add discussions on VisualGPT.
 - (ErrM) Section 3.1: Add bold headings and revise the introduction on the tagger to make it clearer.
 - (ErrM) Section 3.3: Add more intuition on what sentences are removed.
 - (b7uK) Section 5.1 Footnote 2: Remove the claim on the data leakage is not a concern.
 - (ErrM) Section 5.1: Add more details on why selecting $\tau=0.8$.
 - (b7uK) Section 5.3, Table 2: Add ablation study with CLIP model.
 - (b7uK) Section 6: Add more experiment details for the synthetic data experiment.
 - (b7uK) Section 6: Add an example with scene text.
 - (b7uK, ErrM) Appendix A Figure 7: Add a few examples of the prompts for language model.
 - (ErrM) Appendix D: Add discussions on why Socratic model cannot perform well.
 - (8CQ9) Appendix D: Add why it is unreasonable to compare our framework to traditional IPC methods.
 - (ErrM, 8CQ9) Appendix D: Add discussions on the computation cost.
 - (WzAU): Appendix F: Add experiment results on VQA tasks.

---

### Meta-Review · Area_Chair_oo5u · 2022-08-24

**Recommendation:** Accept
**Confidence:** Less certain

**Metareview:**

All three reviewers have voted weak accept to this paper; the authors have engaged well with the reviewers and have improved their paper. I also recommend acceptance.

**Award:**

No

---

### Decision · Program_Chairs · 2022-09-14

Accept